# Sleeping pattern and activities of daily living modulate protein expression in AMD

**Kaushal Sharma**[1,2]**, Ramandeep Singh**[3]**, Suresh Kumar Sharma**[4]**, Akshay Anand**[1]*

**1** Neuroscience Research Lab, Department of Neurology, Post Graduate Institute of Medical Education and Research, Chandigarh, India, **2** Advanced Pediatrics Centre, Department of Pediatrics, Post Graduate Institute of Medical Education and Research, Chandigarh, India, **3** Department of Ophthalmology, Post Graduate Institute of Medical Education and Research, Chandigarh, India, **4** Department of Statistics, Panjab University, Chandigarh, India

* akshay1anand@rediffmail.com

**Data Availability Statement:** All relevant data are in the paper and its Supporting Information files.

**Funding:** This study was funded by the Department of Biotechnology, New Delhi, India to AA (No. BT/PR17550/MED/30/1755/2016). We also

## Abstract

Degeneration of macular photoreceptors is a prominent characteristic of age-related macular degeneration (AMD) which leads to devastating and irreversible vision loss in the elderly population. In this exploratory study, the contribution of environmental factors on the progression of AMD pathology by probing the expression of candidate proteins was analyzed. Four hundred and sixty four participants were recruited in the study comprising of AMD (n = 277) and controls (n = 187). Genetics related data was analyzed to demonstrate the activities of daily living (ADL) by using regression analysis and statistical modeling, including contrast estimate, multinomial regression analysis in AMD progression. Regression analysis revealed contribution of smoking, alcohol, and sleeping hours on AMD by altered expression of *IER-3*, *HTRA1*, *B3GALTL*, *LIPC* and *TIMP3* as compared to normal levels. Contrast estimate supports the gender polarization phenomenon in AMD by significant decreased expression of *SLC16A8* and *LIPC* in control population which was found to be unaltered in AMD patients. The smoking, food habits and duration of night sleeping hours also contributed in AMD progression as evident from multinomial regression analysis. Predicted model (prediction estimate = 86.7%) also indicated the crucial role of night sleeping hours along with the decreased expression of TIMP-3, IER3 and SLC16A8. Results revealed an unambiguous role of environmental factors in AMD progression mediated by various regulatory proteins which might result in intermittent AMD phenotypes and possibly influence the outcome of anti-VEGF treatment.

## 1. Introduction

Most of the degenerative diseases (*e.g.* AMD and Alzheimer's disease) have shown complex phenotypes based on both gene-environment interactions which have propensity to alter the cellular functions by gene expression changes [1, 2]. AMD is characterized by degenerative changes in macular photoreceptors and vision impairment in elderly. It is associated with various environmental factors and 52 independent genetic loci [3]. However, most of reported AMD alleles have not been probed for interaction with environmental factors rendering the genetic studies of AMD an incomplete and unimpactful analysis.

acknowledge CSIR-UGC, New Delhi for providing fellowship during PhD to KS, Department of Science and Technology (DST), New Delhi, India and Indian Council of Medical Research (ICMR), New Delhi, India to provide the travel funds to KS. The funders had no role in study design, data collection and analysis, decision to publish, or preparation of the manuscript.

**Competing interests:** The authors have declared that no competing interests exist.

AMD literature is replete with evidence in support of the contribution of both genetic and environmental factors in the progression of AMD, but fails to define the architecture of this complex phenotype. However, smoking has been much investigated with context to AMD and found to exhibit its effect through generation of oxidative stress [4] and induce angiogenic cascade [5, 6] in order to promote angiogenesis of choroidal blood vessels. Moreover, smoking exposure has been shown to exert the pathological changes akin to AMD by blocking alternative complement pathways and by lipid dysregulation in RPE cells [7]. Studies have also shown that the combined effect of both alcohol consumption and smoking might further exacerbate the AMD pathology by influencing the activity of SOD (superoxide dismutase) and glutathione peroxidase activity [8]. Our previous reports have also defined the pathological role of oxidative stress [9], impaired angiogenesis [10, 11] and inflammatory cascade (mediated through CCL2 and CCR3) [12–14]. Similar pathological hallmarks have also been exhibited by other degenerative diseases including AD, ALS etc. [15–18]. Recently, we have also identified genetic variants of TIMP3, APOE and HTRA1 genes to contribute towards the complexity of Indian AMD [19]. The exact mechanism of action of the associated environmental factors to modulate the function wide genetic architecture in AMD is not adequately investigated although it is generally accepted to play a key role in AMD pathology. Exposure of environmental factors is possibly to bring the epigenetic modifications at the gene/genome (methylation of CpG Island) as well as on histone protein (*acetylation*, *phosphorylation*, *methylation*, *citrullination*, *ubiquitylation*, *ribosylation*, and *sumoylation*) levels which could modulate the expression of proteins and their mediated cellular mechanism [1]. Temporal nature of smoking and dietary induced AMD pathology by altering the protein expression indicates the epigenetic regulation of disease progression [20]. Revealing the understanding of rare and common genetic variants, copy number variations along with mitochondrial genetics, and their contributions in the AMD pathology under the influence of environmental factors, enable us to redefine the diagnosis and propose a new therapeutics regimen [21–23].

We report that there is an alteration in expressions of HtrA Serine Peptidase 1 (*HTRA1*), Tissue inhibitor of metalloproteinase-3 (*TIMP*-3) and Immediate Early Response 3 (*IER*-3) in sleep deprived individuals or AMD patients with increase in sleep duration, prompting further research [24, 25]. This has implication for superior diagnosis and management of patients affected by AMD. We wanted to examine the nature and extent of the role of environmental factors in exerting its influence on genetic components and whether these are governed by epigenetic or epistatic interactions.

## 2. Materials and methods

### 2.1. Recruitment of participants

We have recruited around 464 participants in present study which comprised with both AMD (n = 277) and controls (n = 187). Participants were recruited as per the inclusion-exclusion criteria mentioned in the study along with their informed written consent. The study has started the recruitment of participants from 2010 and finished the same in 2017. The recruitment of participants and clinical examinations were performed from the Department of Ophthalmology, PGIMER, Chandigarh, India and experiments were conducted in Neuroscience Research Lab, PGIMER, Chandigarh, India. The study was conducted as per the approval of Institute Ethical Committee of both PGIMER (No: PGI/IEC/2005-06; dated: 23.07.2013) and Panjab University (IEC No. 131A-1, dated: 29.10.2014). All methods pertaining to study were performed in accordance with the relevant guidelines and regulations laid down by IECs. The study could be considered as a representative to replicate the same in large cohort.

## 2.2. Clinical investigations

Clinical evaluation of AMD phenotypes was carried out by retina specialists which included fluorescein fundus angiography (FFA) of dilated retina of the patients. Patients were clinically classified based on the drusen deposits and leaky vessels captured as fundus images. Moreover, the extent of macular photoreceptors degeneration and thickness of retinal layers were also examined by Optical Coherence Tomography (OCT) of patients. AMD patients were classified based on the clinical features observed and stratified by the AREDS criteria. Based on the presence of clinical parameters including drusen, neovascular lesions and atrophy of photoreceptors, AREDS stratified the AMD patients into 5 categories. Briefly, AMD patients with A few small drusen (<63 microns in diameter) fall in the AREDS 1. AREDS2 was characterized as multiple small drusen, a few intermediate drusen (63 to 124 microns in diameter), and/or RPE abnormalities. Many intermediate drusen with at least one large druse (≥125 microns in diameter) classified as AREDS3. Atrophy of foveal photoreceptors was characterized as greographic atrophy (AREDS4) and finally, patients with any leakage between retinal layers or neovascular features were classified as AREDS5.

## 2.3. Activities for daily living (ADL) details

A well-defined questionnaire was introduced to collect the socio-demographic details of the studied participants. ADL details prominently included the daily living activities (food habits, smoking, alcohol), education and profession, any medication, physical activities and/or yogic practices, sleeping pattern, etc which mostly associated with person's life style. Food habit was categorized *i.e.* vegetarian, prior history of non-vegetarian and/or non-vegetarian, based on the consumption of food for at least six months or more since the date of recruitment. Non-vegetarian participant was defined based on consumption of chicken, fish and/or mutton or any one of them. Smokers were also categorized (non-smoker, prior/past-smoker, current smoker) based on the current and/or past-history of smoking, if any, of the participants who must be smoking for minimum six months (in case of prior or current smoker) at the time of recruitment. Similarly, participants were also classified (non-alcoholic, prior/past-alcoholic and current alcoholic) based on the alcohol consumption (past or current) with minimum 6 months of alcohol consumption history. To see the impact of sleep hours of the participants, we have classified participants in to three categories namely as sleep deprived (<6 hours sleep), 6–7 hours' sleep and rise before 6AM (6–7 hours' sleep) and >6–7 hours' sleep and late sleep or late rise (after 6AM). Moreover, we also have asked participants whether they have been instructed to take medication for any ailment including cardiovascular, hypertension, diabetes, migraine and stroke history by a physician in addition to AMD.

## 2.4. Serum extraction

3ml of blood sample was withdrawn from participants and were subjected to centrifugation at 2500rpm for 30minutes. Upper supernatant layer was collected and stored at -80°C for further experimentation.

## 2.5. Total protein estimation

Total protein in the serum of participants was estimated by Bradford's method. Samples were diluted (ranges from 200–600 times) with distilled water before performing the assay. Bradford's reagent was added in 1:4 dilutions in the experiment. Absorbance of samples was taken at 595nm by ELISA reader (BioRad, USA).

## 2.6. ELISA

Serum levels of proteins involved in pro-angiogenesis (*e.g. ADAMTS9*, *TIMP*-3), cellular regulatory proteins (like *IER*-3, *B3GALTL*, *HTRA*1), monocarboxylic acid (pyruvic acid or lactate) transporter (SLC16A8) and lipid metabolizing proteins [hepatic lipase (*LIPC)* and apolipoprotein E (*APOE*)] were estimated using commercially available ELISA kit. Protocol was followed as per available recommendations with the kit and absorbance was recorded at 450nm. Values of ELISA were normalized with total protein of the respective sample. Levels of protein were compared with control populations.

## 2.7. Genotype analysis

Genotype analysis was also carried out for same set of genes involved in various cellular functions *e.g.* lipid metabolizing proteins [*LIPC* (rs920915) and *APOE* (*rs769449*)], pro-angiogenesis [*e.g. ADAMTS9* (rs6795735), *TIMP*-3 (rs5749482)], cellular regulatory proteins [like *IER*-3 (rs3130783), *B3GALTL* (rs9542236), *HTRA*1 (rs11200638)] and monocarboxylic acid transporter [e.g. SLC16A8 (rs8135665)] to associate with ADL.

## 2.8. Statistics

Data was assessed for normal distribution in the population using normal quantile plot (O-Q plot) and Kolmogorov-Smirnos (K-S) tests. Differential protein expression in various groups stratified on the basis of socio-demographic details, was analyzed by ANOVA. Logistic regression analysis was utilized to analyse the effect of exposure of environmental factors (like smoking, food habit, alcohol consumption *etc*) by creating variables for prior and current status of activities of daily living (ADL). To examine the differential protein expression due to gender polarization effect in AMD patients, contrast analysis was carried out. Predictive modeling based on clinical severity and associated expression changes were analyzed by regression analysis. Multinomial regression analysis was done to analyze the contribution of ADL in AMD severity. Moreover, the prediction model based on ADL and expression level of proteins was put forwarded to diagnose AMD cases more precisely.

# 3. Results

## 3. 1. Association of socio-demographic factors

*Chi-square* analysis of the data revealed a significant association of various factors with AMD patients including profession, accident, consumption of anti-inflammatory drugs of participants. There is a significant difference between mean age of AMD and Control (p<0.001). Results reveal marginal association of physical activity and education of an individual with AMD pathology (Table 1).

Activities of daily living (ADL) of the participants were also analyzed to examine if association existed between AMD and these variables. Association of AMD patients with BMI, smoking habits (both prior and current habit) and abnormal sleeping pattern was noted. Moreover, it was higher in AMD patients as compared to control (Table 2).

Frequencies of clinical features of AMD patients were also calculated as presented in Table 3. Recruited AMD patients showed advanced form of AMD clinical features (AREDS 5) involving bilateral phenotype. Further dissection of bilateral phenotypes of AMD patients revealed the numbers as 28, 34 and 82 with bilateral dry, bilateral wet and dry-wet bilateral phenotypes, respectively. Approximately, 42% of AMD patients were also diagnosed with and cataract and underwent the surgery to treat the same.

**Table 1. Comparative demographic characteristics of AMD and controls.**

| Features | AMD(n) | Controls(n) | p-value |
|---|---|---|---|
| **Gender** | | | |
| 1. Male | 171 (61.73%) | 99 (53.51%) | 0.833 |
| 2. Female | 106 (38.27) | 86 (46.87%) | |
| **Age (Mean ± SD)** | 68.30 ± 9.086 | 56.94 ± 11.266 | <0.0001*** |
| **Anti-Inflammatory drugs¥** | | | <0.0001*** |
| 1. No Inflammatory | 144 (53.33%) | 139 (81.76%) | |
| 2. Anti-Inflammatory drugs | 126 (46.67%) | 31 (18.24%) | |
| **Occupation¥** | | | <0.0001*** |
| 1. Professional | 62 (22.63%) | 8 (5.19%) | |
| 2. Semi professional | 48 (17.52%) | 6 (3.90%) | |
| 3. Clerical | 41 (14.96%) | 37 (24.03%) | |
| 4. Skilled | 07 (2.56%) | 13 (8.44%) | |
| 5. Semi-skilled | 12 (4.38%) | 28 (18.18%) | |
| 6. Unskilled | 103 (37.59%) | 62 (40.26%) | |
| 7. Unemployed | 01 (0.36%) | 0 | |
| **Education¥** | | | 0.063 |
| 1. Professional or honor | 61 (22.18%) | 46 (28.57%) | |
| 2. Graduate or Post Graduate | 21 (7.64%) | 20 (12.42%) | |
| 3. Intermediate | 23 (8.36%) | 18 (11.18%) | |
| 4. High school | 74 (26.91%) | 35 (21.74%) | |
| 5. Middle school | 19 (6.91%) | 15 (9.32%) | |
| 6. Primary school | 57 (20.73%) | 19 (11.80%) | |
| 7. Illiterate | 20 (7.27%) | 08 (4.97%) | |
| **Physical activity¥** | | | 0.052 |
| 1. Physically active | 111 (40.81%) | 78 (49.37%) | |
| 2. Inactivity | 161 (59.19%) | 80 (50.63%) | |
| **Accident history¥** | | | 0.029* |
| 1. Accident history | 55 (19.93%) | 18 (11.69%) | |
| 2. No accident history | 221 (80.07%) | 136 (88.31%) | |

¥ Some missing values.

## 3.2. Activities of daily living influence protein expression

We also attempted to study the gross impact of various ADL on protein expressions in AMD patients. Our results revealed a significantly enhanced LIPC levels in AMD patients who smoke and have non-vegetarian food habits (prior) suggesting an impaired lipid metabolism (IDL to LDL formation) due to malfunction of LIPC in AMD pathology (Fig 1A & 1E). Interestingly, the sleeping pattern of AMD patients [6-7hrs sleep, waking time before 6AM in morning (normal sleep) versus >7-8hrs sleep, late sleep or late wakefulness] was found to display a significant effect on HTRA1 levels. Documentation of consequently altered HTRA1 levels suggests the role of impaired circadian rhythm on AMD patients and the biological significance of HTRA1 being amenable to such regulation. However, more research is required (Fig 1G). We did not find significant alteration in protein levels under the influence of smoking, participant's food habits and disturbed sleeping pattern (Fig 1B–1D, 1F and 1H).

The *beta* coefficient (B) of logistic regression analysis revealed that significantly decreased expression of IER-3 (-0.288), B3GALTL (-0.214), LIPC (-0.172), TIMP-3 (-63.696) along with increased levels of HTRA1 (0.696) were observed in Indian AMD, without adjusting the ADL.

**Table 2. Comparative frequencies of activities of daily livings (like BMI, smoking, alcohol consumption, food habit and sleeping pattern) of AMD and control participants.**

| Features | AMD (n) | Controls (n) | p-value |
|---|---|---|---|
| **BMI¥** | | | 0.003* |
| 1. **Under weight** | 10 (3.75%) | 07 (4.46%) | |
| 2. **Normal** | 175 (65.54%) | 87 (55.41%) | |
| 3. **Over Weight** | 50 (18.73%) | 53 (33.76%) | |
| 4. **Obese** | 32 (11.98%) | 10 (6.37%) | |
| **Smoking habit¥** | | | 0.010* |
| 1. **Never smoker** | 185 (67.52%) | 128 (81.01%) | |
| 2. **Prior smoker** | 54 (19.71%) | 17 (10.76%) | |
| 3. **Current smoker** | 35 (12.77%) | 13 (8.23%) | |
| **Alcohol consumption** | | | 0.650 |
| 1. **Never Alcohol** | 186 (67.15%) | 112 (71.34%) | |
| 2. **Prior Alcohol** | 30 (10.83%) | 14 (8.92%) | |
| 3. **Current Alcohol** | 61 (22.02%) | 31 (19.74%) | |
| **Food habit ¥** | | | 0.163 |
| 1. Vegetarian | 147 (53.26%) | 78 (50%) | |
| 2. Non-vegetarian | 86 (31.16%) | 61 (39.10%) | |
| 3. Prior nonveg | 43 (15.58%) | 17 (10.90%) | |
| **Night sleeping hours¥ 269** | | | 0.006* |
| 1. **6–7 hrs sleep, rise before 6AM** | 157 (58.36%) | 81 (54.36%) | |
| 2. **Sleep deprived (<6hrs sleep)** | 29 (10.78%) | 05 (3.35%) | |
| 3. **>7–8 sleep, late sleep or late rise (after 6AM)** | 83 (30.86%) | 63 (42.29%) | |

¥ Some missing values.

Logistic regression analysis estimated the individual effect of either prior or current status of ADL on protein expressions in AMD pathology (Table 4). Similar results were noted by adjusting smoking and alcohol habits. Past history of alcohol consumption was also found to significantly decrease IER3, B3GALTL, LIPC, TIMP3 expressions and increase HTRA1 levels. Additionally, prior history of alcohol consumption has potential to modulate the AMD pathology by -0.641 unit as compared to those who consume vegetarian diet (95% CI = 0.278–0.998; P = 0.049). Prior non-vegetarian history revealed marginal association with AMD by

**Table 3. Clinical characteristics of North-West Indian AMD recruited in the study.**

| AMD features | Phenotypes | Number | Percent (%) |
|---|---|---|---|
| **AMD phenotypes** | Dry AMD | 42 | 15.2 |
| | Wet AMD | 91 | 32.9 |
| | Bilateral AMD | 144 | 52.0 |
| **Cataract¥** | No cataract | 157 | 57.30 |
| | Unilateral Cataract | 53 | 19.34 |
| | Bilateral cataract | 64 | 23.36 |
| **Eye surgery¥** | No eye surgery | 160 | 64.78 |
| | Unilateral surgery | 96 | 35.03 |
| | Bilateral surgery | 18 | 06.57 |

¥ Some missing values.

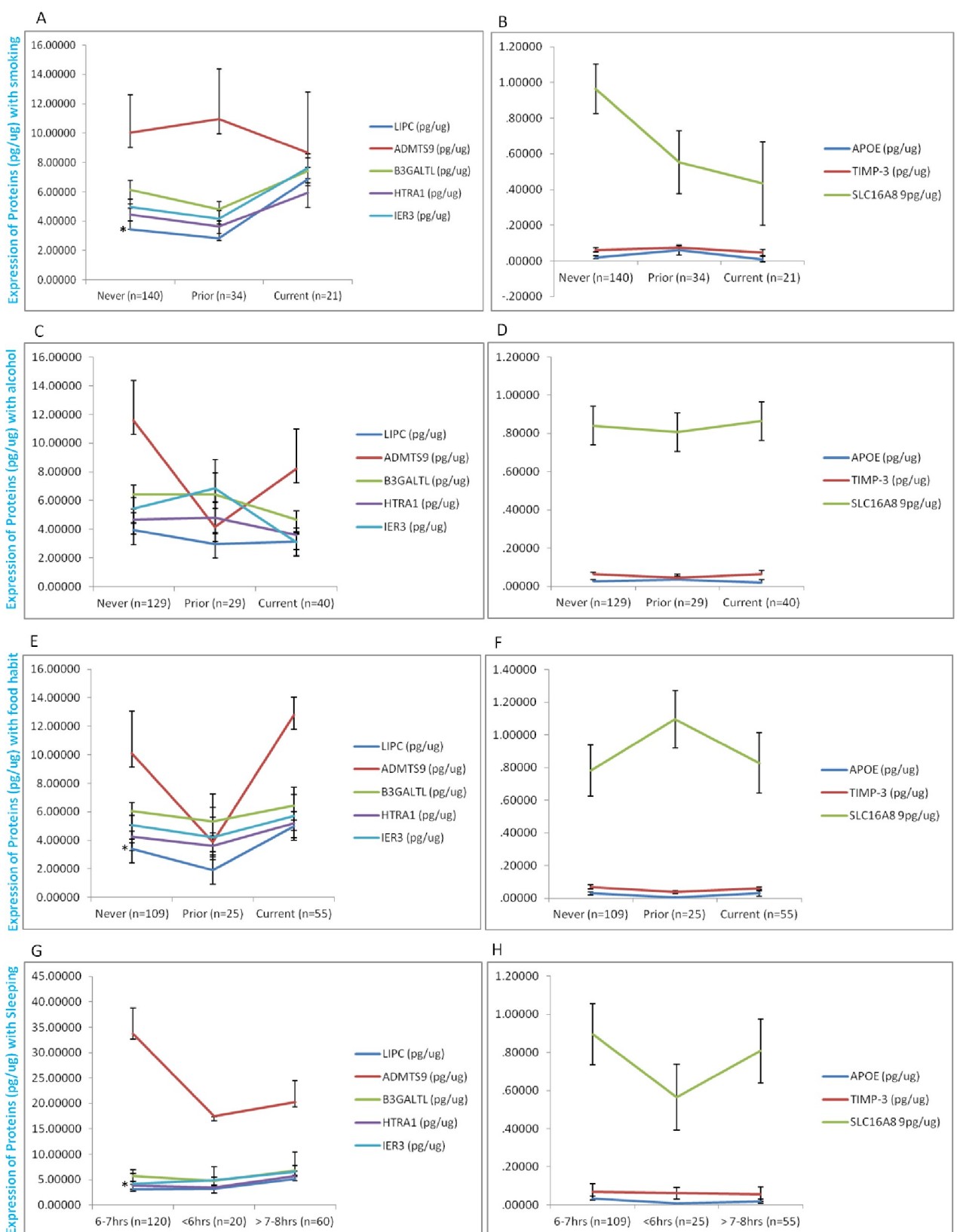

**Fig 1. Expression of protein under ADL.** Expression of lipid metabolizing (LIPC and APOE), proagniogenic (TIMP3 and ADAMTS9), regulatory (HTRA1, B3GALTL, and IER3) and momocarboxylic acid transporter (SLC16A8) proteins under the influence of ADL. LIPC (pg/ug) levels were

significantly elevated in 'current smoker' AMD patients (A & E). Altered sleeping patter can be associated with HTRA1 levels in AMD pathology indicating the crucial role of circadian rhythm in degenerative diseases like AMD (G).

modulating the expression of HTRA, B3GALTL, IER3, LIPC and TIMP3 with changing it -0.691 (B = beta coefficient) unit with reference to never smoker (95% CI = 0.233–1.076; P = 0.076). Interestingly, by adjusting the sleeping pattern of AMD patients, it decreased the expression of IER3 (B = -.351; 95% CI = .605-.819; P = <0.0001) and TIMP3 (B = -44.128; P = <0.0001) significantly. Moreover, altered sleeping pattern (person who slept late or woke up after sun rise) revealed the changes in the expression of IER3 and TIMP3 by 0.757 unit (B coefficient) as compared to normal sleep (95% CI = 1.2–3.785; P = 0.01). Significant changes in IER3 (B = -.314; 95% CI = .637-.838; P = <0.0001), TIMP3 (B = -41.969; P = <0.0001) and SLC16A8 (B = -0.184; 95% CI = 1.022–1.415; P = 0.027) expression were observed while adjusting the physical activity of AMD patients (Table 4). Our results support the previous findings which indicate the crucial contribution of environmental factors including smoking, food habits, physical activity and alcohol consumption in AMD pathology by regulating the proteins expression. The association of sleeping pattern with AMD shows the biological importance of HTRA1, IER3 and TIMP3 which may have roles in modulating age-related changes in retinal layers, representative of AMD pathology.

### 3.3. Gender polarization effects of SLC16A8 and LIPC expressions in AMD

Females are considered to be more susceptible for AMD pathology, though we did not find any significant difference in frequency of AMD male and female. We also attempted to examine the gender effect on protein expression in Indian AMD patients. Contrast estimate was done using univariate analysis of variance to analyze the difference in protein expressions among male and female of studied population (Table 5). Contrast estimate (CE) for SLC16A8 [CE = -0.768; F = 5.451; 95% CI = -1.418- (-)-0.119; P = 0.021] and LIPC [CE = -0.644; F = 7.357; 95% CI = -1.112- (-)-0.175; P = 0.007] was found to be significantly decreased between male and female control population. Such differential expression of both proteins was not observed among AMD male and female. It may be argued that differential expression of both SLC16A8 and LIPC is required to regulate various mechanisms under the influence of a set of hormones and may confer the protective mechanism for age-related changes.

### 3.4. ADL contribution in advancement of AMD severity

To assess the independent contribution of ADL (including smoking, food habits, physical activity, sleeping hours and alcohol consumption) on AMD severity (AREDS criteria), we subjected the data to multinomial logistic regression. The model demonstrated a highly significant association of both past (B = -1.286; P = <0.0001) and current non-vegetarian food habit (B = -0.667; P = 0.001) in the advancement of AMD pathology (Table 6). Results showed that current non-vegetarian and past history of non-vegetarian history can contribute to AMD by B values of -0.667 and -1.286 units as compared to reference category (vegetarian diet). However, the prediction of model was not satisfactory.

Similarly, past and current status of smoking has also showed a significant association in progression of AMD pathology. Contribution of past (B = -1.275; P = <0.0001) and current smoking (B = -2.435; P = <0.0001) was observed in exacerbating the AMD pathology with prediction probability of around 68.4% (Table 6). Results for alcohol consumption in progression of AMD pathology has shown a comparable trend highlighting the contribution of past (B = -1.803; P = <0.0001) and current status of alcohol consumption (B = -1.077; P = <0.0001) as

**Table 4. Logistic regression analysis to estimate the changes in protein expression under the influence of ADL.**

| | | | | | | | | 95% C.I. | |
|---|---|---|---|---|---|---|---|---|---|
| | | | | | | **Variables in the Equation** | | | |
| | | **B** | **S.E.** | **Wald** | **Df** | **Sig.** | **Unadjusted** | **Lower** | **Upper** |
| **Step 5e** | IER3 levels | -.288 | .066 | 19.371 | 1 | **<0.0001** | .749 | .659 | .852 |
| | B3GALTL levels | -.214 | .065 | 11.037 | 1 | **0.001** | .807 | .711 | .916 |
| | HTRA1 levels | .696 | .149 | 21.744 | 1 | **<0.0001** | 2.006 | 1.497 | 2.687 |
| | LIPC levels | -.172 | .081 | 4.539 | 1 | **0.033** | .842 | .719 | .986 |
| | TIMP3 levels | -63.696 | 8.666 | 54.027 | 1 | **<0.0001** | .000 | .000 | .000 |
| | Constant | 1.484 | .220 | 45.353 | 1 | 0.000 | 4.412 | | |
| | | **B** | **S.E.** | **Wald** | **df** | **Sig.** | **Adjusted for smoking** | **95% C.I.** | |
| | | | | | | | | **Lower** | **Upper** |
| **Step 1a** | IER3 levels | -.287 | .066 | 18.980 | 1 | **<0.0001** | .751 | .660 | .854 |
| | B3GALTL levels | -.214 | .064 | 11.335 | 1 | **0.001** | .807 | .713 | .914 |
| | HTRA1 levels | .690 | .147 | 21.984 | 1 | **<0.0001** | 1.994 | 1.494 | 2.660 |
| | LIPC levels | -.171 | .082 | 4.345 | 1 | **0.037** | .843 | .717 | .990 |
| | TIMP3 levels | -62.852 | 8.660 | 52.672 | 1 | **<0.0001** | .000 | .000 | .000 |
| | Smoking code | | | 1.649 | 2 | 0.438 | | | |
| | Smoking code(1) | -.507 | .395 | 1.648 | 1 | 0.199 | .602 | .278 | 1.306 |
| | Smoking code(2) | -.049 | .494 | .010 | 1 | 0.920 | .952 | .361 | 2.508 |
| | Constant | 1.538 | .226 | 46.431 | 1 | 0.000 | 4.657 | | |
| | | **B** | **S.E.** | **Wald** | **df** | **Sig.** | **Adjusted for Alcohol** | **95% C.I.** | |
| | | | | | | | | **Lower** | **Upper** |
| **Step 1a** | IER3 levels | -.291 | .068 | 18.513 | 1 | **<0.0001** | .748 | .655 | .854 |
| | B3GALTL levels | -.222 | .065 | 11.756 | 1 | **0.001** | .801 | .705 | .909 |
| | HTRA1 levels | .707 | .151 | 21.784 | 1 | **<0.0001** | 2.028 | 1.507 | 2.728 |
| | LIPC levels | -.185 | .083 | 4.983 | 1 | **0.026** | .831 | .706 | .978 |
| | TIMP3 levels | -63.702 | 8.623 | 54.572 | 1 | **<0.0001** | .000 | .000 | .000 |
| | Alc code | | | 5.397 | 2 | 0.067 | | | |
| | Alc code(1) | -.679 | .441 | 2.374 | 1 | 0.123 | .507 | .214 | 1.203 |
| | Alc code(2) | -.641 | .326 | 3.870 | 1 | **0.049** | .527 | .278 | .998 |
| | Constant | 1.705 | .246 | 47.879 | 1 | 0.000 | 5.502 | | |
| | | **B** | **S.E.** | **Wald** | **df** | **Sig.** | **Adjusted for Food habit** | **95% C.I.** | |
| | | | | | | | | **Lower** | **Upper** |
| **Step 1a** | IER3 levels | -.286 | .067 | 18.201 | 1 | **<0.0001** | .751 | .659 | .857 |
| | B3GALTL levels | -.209 | .065 | 10.368 | 1 | **.001** | .811 | .714 | .921 |
| | HTRA1 levels | .674 | .149 | 20.503 | 1 | **.000** | 1.963 | 1.466 | 2.628 |
| | LIPC levels | -.163 | .081 | 4.017 | 1 | **.045** | .850 | .724 | .996 |
| | TIMP3 levels | -63.781 | 8.722 | 53.475 | 1 | **<0.0001** | .000 | .000 | .000 |
| | Food Habit code | | | 3.219 | 2 | .200 | | | |
| | Food Habit code(1) | -.058 | .286 | .040 | 1 | .841 | .944 | .539 | 1.655 |
| | FoodHabit_code(2) | -.691 | .390 | 3.140 | 1 | .076 | .501 | .233 | 1.076 |
| | Constant | 1.613 | .255 | 40.018 | 1 | .000 | 5.018 | | |
| | | **B** | **S.E.** | **Wald** | **df** | **Sig.** | **Adjusted for sleeping** | **95% C.I.** | |
| | | | | | | | | **Lower** | **Upper** |
| **Step 1a** | IER3 levels | -.351 | .077 | 20.720 | 1 | **< .0001** | .704 | .605 | .819 |
| | TIMP3 levels | -44.128 | 7.184 | 37.735 | 1 | **< .0001** | 0.000 | .000 | .000 |
| | Night Slp code | | | 8.606 | 2 | .014 | | | |
| | Night Slp code(1) | -.568 | .616 | .851 | 1 | .356 | .567 | .169 | 1.895 |

*(Continued)*

**Table 4.** (Continued)

| | | | | | | | | |
|---|---|---|---|---|---|---|---|---|
| | | | | | **Variables in the Equation** | | | |
| | Night Slp code(2) | .757 | .293 | 6.666 | 1 | **.010** | 2.131 | 1.200 | 3.785 |
| | Constant | 1.367 | .239 | 32.617 | 1 | .000 | 3.923 | | |

| | | B | S.E. | Wald | df | Sig. | Adjusted for Physi activity | 95% C.I. | |
|---|---|---|---|---|---|---|---|---|---|
| | | | | | | | | Lower | Upper |
| **Step 1a** | IER3 levels | -.314 | .070 | 20.140 | 1 | **< .0001** | .730 | .637 | .838 |
| | TIMP3 levels | -41.969 | 6.908 | 36.913 | 1 | **< .0001** | .000 | .000 | .000 |
| | SLC16A8 levels | .184 | .083 | 4.913 | 1 | **.027** | 1.202 | 1.022 | 1.415 |
| | Physi Activ code(1) | -.039 | .263 | .022 | 1 | .883 | .962 | .574 | 1.611 |
| | Constant | 1.350 | .276 | 23.919 | 1 | .000 | 3.859 | | |

compared to reference category (Table 6). The prediction probability of the model was about 65%. Interestingly, sleep deprived (<6hours sleep) and >7-8hrs sleep, late sleep or late rise subjects have also shown the significant impact on progression of AMD severity. Results have shown that sleep deprived (B = -1.885; P = <0.0001) and >7-8hrs sleep, late sleep or late rise (B = -.681; P = <0.0001) patterns contribute to the progression of AMD severity with a prediction probability of about 60% (Table 6). Pearson and deviance values of Goodness-of-fit model were found to be non-significant for the analysis. Results are suggested an independent role of ADL (environmental factors), especially sleep, in the progression of AMD pathology-which has never been analyzed previously.

### 3.5. Altered sleeping pattern and expression of IER3, TIMP3 and SLC16A8 confer the AMD

Association of sleep pattern and AMD pathology has not been adequately investigated. We have attempted to further dissect the impact of sleeping pattern in AMD patients. Regression analysis shows that night sleeping hours (B = 0.449; Exp(B) = 1.567; 95% CI = 1.1–2.23; P = 0.013) along with the expression of IER3 (B = -.444; Exp(B) = 0.641; 95% CI = 0.512–0.804; P = <0.0001) and TIMP3 (B = -23.54; Exp(B) = <0.0001; 95% CI = 0.000–0.004; P = 0.010) are significantly associated with AMD pathology. However, the marginal association of SLC16A8 expression (B = -.332; Exp(B) = .717; 95% CI = 0.506–1.017; P = 0.062) was also observed (Table 7). Results suggest the imperative role of sleeping pattern of an individual which may activate the various age-related mechanisms by influencing pertaining protein expressions. Our results indicate the biological significance of IER3, TIMP-3 and SLC16A8 expression to be influenced by alter sleeping hours of an individual. Classification table also strengthens our hypothesis with 86.7% validity of this regression model to predict the AMD pathology.

**Table 5. Contrast estimate using univariate analysis of variance to differentiate the expression pattern on basis of gender for control population.**

| Variables | F-value | Contrast estimate (CE) | p-value | 95% CI | |
|---|---|---|---|---|---|
| | | | | Lower | Higher |
| **SLC16A8** | 5.451 | -0.768 | 0.021 | -1.418 | -0.119 |
| **LIPC** | 7.357 | -0.644 | 0.007 | -1.112 | -0.175 |

[Female (n) = 86; male (n) = 99].

**Table 6. Multinomial logistic regression to examine the contribution of ADL in AMD severity.**

| | Parameters | B | S.E. | Wald | df | p-value |
|---|---|---|---|---|---|---|
| | | | Parameter estimates | | | |
| **Food habit** | Non-vegetarian | -0.667 | .198 | 11.388 | 1 | **0.001** |
| | Prior Non-vegetarian | -1.286 | .247 | 27.020 | 1 | **<0.0001** |
| **Smoking** | Past smoker | -1.275 | .288 | 19.572 | 1 | **<0.0001** |
| | Current smoker | -2.435 | .390 | 39.054 | 1 | **<0.0001** |
| **Night Sleep hours** | Sleep deprived (<6hrs sleep) | -1.885 | .311 | 36.716 | 1 | **<0.0001** |
| | >7-8hrs sleep, late sleep or late rise | -.681 | .195 | 12.206 | 1 | **<0.0001** |
| **Alcohol** | Past Alcohol | -1.803 | .280 | 41.547 | 1 | **<0.0001** |
| | Current Alcohol | -1.077 | .209 | 26.619 | 1 | **<0.0001** |

Reference category: [a]Vegetarian habit; [a]non-smoker habit; [a]6-7hours sleep or wake up before 6AM; [a]Never alcoholic.

## 4. Discussion

Disease pathology of AMD is known to be influenced by both genetic and environmental factors evident by our quantitative outcome of protein expression under the influence of environmental factors [1]. In general, the ambiguity in the nature and extent of interaction between environmental and genetic factors has significantly hampered the pace of clinical translation in the field of AMD genetics [2]. Current AMD genetics warrants comprehensive analysis in the manner it can illustrate the contribution and interactions of contributory factors along with their degree of penetrance in disease progression. Majority of ageing diseases progress by cumulative genetic changes under temporal exposure of ADL consequently result in cellular and molecular alterations including protein homeostasis, metabolic dysfunction and aberrant signaling processes. The altered cellular and molecular crosstalk may confer complexity to the age related diseases thereby confounding an effective and precise diagnosis and treatment regime for complex disorders like AMD [26]. Therefore, a careful consideration of environmental and genetic components and their nature of interactions (and/or extent of interaction) may likely provide a precise AMD phenotype and personalized management strategies. The treatment strategy which can deal with the synergistic and/or cumulative action of contributory factors could provide a better outcome to therapies for AMD [26, 27]. Our ANOVA

**Table 7. Regression analysis to predict the AMD pathology under the influence of ADL.**

| | B | S.E. | Wald | Df | p-value | Exp(B) | 95% CI |
|---|---|---|---|---|---|---|---|
| | | | Variables in the Equation | | | | |
| **Night sleep pattern** | .449 | .180 | 6.205 | 1 | **0.013** | 1.567 | 1.1–2.23 |
| **TIMP3 levels** | -23.54 | 9.194 | 6.555 | 1 | **0.010** | .000 | 0.000–0.004 |
| **IER3 levels** | -.444 | .115 | 14.907 | 1 | **<0.0001** | .641 | 0.512–0.804 |
| **SLC16A8 levels** | -.332 | .178 | 3.487 | 1 | **0.062** | .717 | 0.506–1.017 |
| **Constant** | .192 | .445 | .187 | 1 | 0.666 | 1.212 | |

| | | Classification table | | |
|---|---|---|---|---|
| | | | Predicted | |
| | | Group code | | Percentage corrected |
| | | AMD | Controls | |
| **Group** | AMD | 111 | 17 | **86.7** |
| | Control | 22 | 50 | 69.4 |
| **Overall percentage** | | | | 80.5 |

results demonstrate that smoking and non-vegetarian food habit can effectively alter the LIPC expression that may exacerbate the AMD pathology. Interestingly, altered expression of HTRA1 under the influence of altered sleep cycle, can accelerate the AMD pathology thereby providing opportunity to correct the dysregulated circadian rhythm.

Various studies have been carried out to illustrate the significance of environmental factors on genetic components. Our results show that smoking, gender, age, diet *etc* as contributing confounders and have been significantly associated with complement factors, CFH variant, other variants of other genetic loci including *ARMS2*, *IL-8*, *TIMP3*, *SLC16A8*, *RAD51B*, *VEGFA etc* [28–31]. In our earlier report, smoking was found to be associated with TC genotype of CFH variant (Y402H) along with marginal association of AG genotype of TLR3 (rs3775291) with non-vegetarian food habit which also exhibited confounding effect on CFH expression and modulated TLR3 mediated functions in AMD [32, 33]. Interestingly, we also have found the pathological role of TIMP-3, SLC16A8, IER3 and LIPC in CFH independent manner in Indian AMD [34]. Moreover, eotaxin-2 was also significantly altered when smoker and non-smoker AMD cases were compared [35]. These results point out that the interaction between genetic and environmental factors which often lead to complex phenotype of disease [36].

Logistic regression analysis, by creating the dummy variables, enabled us to identify the effect of prior and current status of ADL like smoking and food habits *etc*. The results unambiguously reveal that prior or current history of non-vegetarian diet, smoking and alcohol can significantly alter the expression of IER-3, TIMP-3, B3GALTL, LIPC and HTRA1, suggesting the involvement of prior exposure of these habits as responsible for changes that may activate the age related molecular and cellular mechanisms. However, not many studies have revealed the association and biological significance of sleeping hours on AMD pathology. Khurana *et al* (2016) reported the high chance of geographic atrophy with increase in sleeping hours [24]. Similarly, short sleep has also been reported to be associated with increased susceptibility of AMD [25]. Similarly, our results from regression anlaysis indicate the pathological implication of altered sleeping hours of AMD. This illustrates the mechanistic importance of HTRA1, IER-3 and TIMP-3 in regulation of circadian rhythms. A marginal association was also reported for SLC16A8. Multinomial regression analysis showed a significant contribution of sleeping hours in AMD progression along with existing factors like smoking, alcohol, food habit *etc*. Temporal protein expressions in differential environmental exposure indicate the plausible role of epigenetics in AMD which has been evident by the 48% higher activity of DNA methyltransferases (DNMTs) in addition to enhanced DNMT1 and DNMT3B levels in AMD as compare to controls. Results also showed the higher methylation of LINE1 in AMD patients [37]. Methylation analysis has demonstrated the epigenetic regulation of *SKI*, *GTF2H4*, *TNXB and IL17*RC genes and their mediated functions in AMD pathology [38, 39].

Gender has additionally been found associated with AMD showing higher susceptibility for females ([40]. However, we did not find any significant difference in frequency between AMD females and males. Surprisingly, contrast estimate results showed differential expression of SLC16A8 and LIPC between control male and female (was not seen among AMD male and female) which may support the sex susceptibility and gender polarization hypothesis in the context of ADL. However, hormonal difference between both genders, their different cellular and molecular action, along with association with SLC16A8 and LIPC, has not been investigated in this report.

## 5. Conclusion

Conclusively, consideration of environmental factor, sleeping patterns and genetics of an individual must be profiled in order to provide the precise diagnostic and therapeutic benefit to

AMD patients. Genetic interaction, gene-protein interaction and gene-environmental interaction, along with nature of interactions and investigation of epigenetic pattern, can enable us to understand the penetrance of each component while facilitating personalized medicine hypothesis. Moreover, exploratory studies to examine the multiple genetic variations (especially in heterogenic disease like AMD), the degree of genetic penetrance of 'hot spots' or other genetic variants (mutation penetrance) may develop various genetic recombinant phenotypes (with varied genetic interactions) for disease pathology under the influence of environmental factors [41, 42]. Hence, complete mapping of genetic interactions, their genetic penetrance, epigenetics status and grading of epistatic interactions under the influence of confounder will provide precise disease phenotype. This could be dealt by modulating the therapeutics. Instead of cellular therapies, herbal or natural therapies could provide benefit in environmental induced age related changes or diseases by regulating the cellular and molecular pathways [43–47]. However, this requires an ADL framework for optimal treatment outcome.

## 6. Strengths and limitations

Study has first time demonstrated the biological significance sleeping pattern, in addition to already existing confounders (*e.g.* smoking, food habits, alcohol consumption) in AMD pathology by examining the altered expression of prominent biomarkers. Sleeping pattern could regulate the angiogenesis and survival of photoreceptors in AMD pathology as indicated by results described in Table 4. Moreover, interesting involvement of SLC16A8 and LIPC (Table 5) in protection mechanism has also provided the pilot data for further investigation in field of AMD which suggest further diversification and complexity of AMD to strengthen the diagnostics and therapeutic outcome accordingly [48]. This led hamper the clinical translation in neurodegenerative diseases including Alzheimers disease and AMD [18, 49]. However, further validation and replication of the results must be reconfirmed in larger cohort (by including Asian and Caucasian population) with precise mechanism of AMD pathogenesis.

## Supporting information

**S1 File. Sleeping pattern and activities of daily living modulate protein expression in AMD (PONE-D-20-29337R1).**
(DOCX)

## Acknowledgments

We sincerely acknowledge all the participants who were involved in the study. We are also thankful to staff of Neuroscience Research Lab. We also sincerely acknowledge Dr Pramod Avti, Department of Biophysics, Post Graduate Institute of Medical Education and Research, Chandigarh, India for the further editing of the manuscript.

## Author Contributions

**Conceptualization:** Kaushal Sharma, Akshay Anand.

**Formal analysis:** Kaushal Sharma, Suresh Kumar Sharma.

**Funding acquisition:** Akshay Anand.

**Investigation:** Kaushal Sharma, Ramandeep Singh.

**Methodology:** Kaushal Sharma, Ramandeep Singh.

**Project administration:** Akshay Anand.

**Software:** Kaushal Sharma.

**Supervision:** Suresh Kumar Sharma.

**Writing – original draft:** Kaushal Sharma.

**Writing – review & editing:** Ramandeep Singh, Suresh Kumar Sharma, Akshay Anand.

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
