## [Decision Letter · Decision Letter 0]

6 Nov 2020

PONE-D-20-29337

Sleeping pattern and activities of daily living modulate protein expression in AMD

PLOS ONE

Dear Dr. Anand,

Thank you for submitting your manuscript to PLOS ONE. After careful consideration, we feel that it has merit but does not fully meet PLOS ONE’s publication criteria as it currently stands. Therefore, we invite you to submit a revised version of the manuscript that addresses the points raised during the review process.

Please evaluate the role of epigenetic changes in the onset and progression of AMD.

We look forward to receiving your revised manuscript.

Kind regards,

Alfred S Lewin, Ph.D.

Academic Editor

PLOS ONE

Journal Requirements:

2. In your Methods section, please provide additional information about the participant recruitment method and the demographic details of your participants. Please ensure you have provided sufficient details to replicate the analyses such as: a) the recruitment date range (month and year), b) a description of any inclusion/exclusion criteria that were applied to participant recruitment, c) a table of relevant demographic details, d) a statement as to whether your sample can be considered representative of a larger population, e) a description of how participants were recruited, and f) descriptions of where participants were recruited and where the research took place.

3.We suggest you thoroughly copyedit your manuscript for language usage, spelling, and grammar. If you do not know anyone who can help you do this, you may wish to consider employing a professional scientific editing service.  

4.We note that you have indicated that data from this study are available upon request. PLOS only allows data to be available upon request if there are legal or ethical restrictions on sharing data publicly. For information on unacceptable data access restrictions, please see http://journals.plos.org/plosone/s/data-availability#loc-unacceptable-data-access-restrictions.

5.Thank you for stating the following in the Acknowledgments Section of your manuscript:

[Department of Biotechnology (No. BT/PR17550/MED/30/1755/2016), New Delhi,

India. The funders had no role in study design, data collection and analysis, decision to publish,

or preparation of the manuscript.]

 [The funders had no role in study design, data collection and analysis, decision to publish, or preparation of the manuscript.]

Reviewers' comments:

Reviewer's Responses to Questions

**Comments to the Author**

1. Is the manuscript technically sound, and do the data support the conclusions?

Reviewer #1: Yes

2. Has the statistical analysis been performed appropriately and rigorously? 

Reviewer #1: Yes

3. Have the authors made all data underlying the findings in their manuscript fully available?

Reviewer #1: Yes

4. Is the manuscript presented in an intelligible fashion and written in standard English?

Reviewer #1: Yes

5. Review Comments to the Author

Reviewer #1: The manuscript is interesting and well organized. The methods selected and applied for the analyzes are correct and adequate. As the rationale of the paper evaluates "Sleeping pattern and activities of daily living" it is also crucial to discuss the possible role of epigenetics in the onset and progression of AMD. Numerous papers have been published in the literature describing a possible role of epigenetic mechanisms in the etiopathogenesis of the disease. This topic should be included in both the introduction and discussion with the aim of improving the originality of the manuscript.

To date, when studying multifactorial diseases it is very important to take into account the role of epigenetics.

6. PLOS authors have the option to publish the peer review history of their article (what does this mean?). If published, this will include your full peer review and any attached files.

Reviewer #1: No

---

## [Author Response · Author response to Decision Letter 0]

26 Jan 2021

PONE-D-20-29337

Sleeping pattern and activities of daily living modulate protein expression in AMD

Dear Editor

We have addressed all the comments raised by the reviewers and modified the manuscript as per journal requirements. Please see revised manuscript for your kind consideration. 

Please evaluate the role of epigenetic changes in the onset and progression of AMD.

Response: We have added the requisite literature to support our results of the manuscript. Thanks for valuable comment. 

Sincerely

Akshay Anand, PhD

Professor

Neuroscience Research lab

Department of Neurology

PGIMER, Chandigarh, India.

Journal Requirements:

Response: Thanks for comment. Manuscript has been changed as per the template provided. 

2. In your Methods section, please provide additional information about the participant recruitment method and the demographic details of your participants. Please ensure you have provided sufficient details to replicate the analyses such as: a) the recruitment date range (month and year), b) a description of any inclusion/exclusion criteria that were applied to participant recruitment, c) a table of relevant demographic details, d) a statement as to whether your sample can be considered representative of a larger population, e) a description of how participants were recruited, and f) descriptions of where participants were recruited and where the research took place.

 Response 2: a) the recruitment date range (month and year)

Response: The methodology has been updated as per the reviewer comment. 

b). a description of any inclusion/exclusion criteria that were applied to participant recruitment.

Response: We have expanded the inclusion criteria of the manuscript, based on which we have recruited the participants. 

c) a table of relevant demographic details

Response: Thanks for the comment. We have mentioned the socio-demographic characteristics of the recruited participants in Table 1, 2 and 3. Please see for your kind consideration. 

d) a statement as to whether your sample can be considered representative of a larger population. 

Response: We have added the statement as reviewer suggested. Please see ‘Recruitment of participants’ section of the manuscript. 

e) a description of how participants were recruited.

Response: The recruitment of the participants was done as per the inclusion and exclusion criteria of the study as mentioned in the ‘inclusion criteria and recruitment of participants’ sections of the manuscript. 

f) descriptions of where participants were recruited and where the research took place.

Response: The recruitment of participants and clinical examination were done from Department of Ophthalmology, PGIMER, Chandigarh, India and experiments were conducted in Neuroscience Research Lab, Dept of Neurology, Post Graduate Institute of medical Education and Research, Chandigarh, India. We have updated the recruitment of participants section as per the reviewer comment. 

Response: We have substantially edited the manuscript with native English to enhance the readability of the same. We have also acknowledges the same in the ‘acknowledgment section’ of the manuscript. Thanks for your comment. 

 Response: Dr Pramod Avti, associate Professor, Department of Biophysics, Post Graduate Institute of Medical Education and Research has help us in editing of the manuscript. We are thankful and acknowledged the same in the manuscript. 

 Response: We have already published an article on genetic analysis which is accessible in the pubmed portal without any restriction (doi:10.1016/j.ygeno.2020.09.044). 

Response: Not applicable. 

 Response: We have recently published an article and could be used as data source for current manuscript (doi: 10.1016/j.ygeno.2020.09.044). Meanwhile, we will initiate the process to upload the data in public repository. 

[Department of Biotechnology (No. BT/PR17550/MED/30/1755/2016), New Delhi,

India. The funders had no role in study design, data collection and analysis, decision to publish, or preparation of the manuscript.]

 [The funders had no role in study design, data collection and analysis, decision to publish, or preparation of the manuscript.]

Response: We have removed the funding statement from the manuscript as per the suggestion of the reviewer. 

Response: Please write the funding statement as “Department of Biotechnology (No. BT/PR17550/MED/30/1755/2016), New Delhi, India and no role in study design, data collection and analysis, decision to publish, or preparation of the manuscript”. 

Reviewers' comments:

Reviewer's Responses to Questions

Reviewer #1: The manuscript is interesting and well organized. The methods selected and applied for the analyzes are correct and adequate. As the rationale of the paper evaluates "Sleeping pattern and activities of daily living" it is also crucial to discuss the possible role of epigenetics in the onset and progression of AMD. Numerous papers have been published in the literature describing a possible role of epigenetic mechanisms in the etiopathogenesis of the disease. This topic should be included in both the introduction and discussion with the aim of improving the originality of the manuscript.

To date, when studying multifactorial diseases it is very important to take into account the role of epigenetics.

Response: Thanks for valuable comment. We have updated the manuscript by discussing the significance of epigenetic modification and its implication in genetic and pathological complexity of AMD. Please see introduction and discussion sections of the manuscript.

---

## [Decision Letter · Decision Letter 1]

1 Mar 2021

Sleeping pattern and activities of daily living modulate protein expression in AMD

PONE-D-20-29337R1

Dear Dr. Anand,

We’re pleased to inform you that your manuscript has been judged scientifically suitable for publication and will be formally accepted for publication once it meets all outstanding technical requirements.

Kind regards,

Alfred S Lewin, Ph.D.

Section Editor

PLOS ONE

Additional Editor Comments (optional):

Reviewers' comments:

Reviewer's Responses to Questions

**Comments to the Author**

1. If the authors have adequately addressed your comments raised in a previous round of review and you feel that this manuscript is now acceptable for publication, you may indicate that here to bypass the “Comments to the Author” section, enter your conflict of interest statement in the “Confidential to Editor” section, and submit your "Accept" recommendation.

Reviewer #1: All comments have been addressed

2. Is the manuscript technically sound, and do the data support the conclusions?

Reviewer #1: Yes

3. Has the statistical analysis been performed appropriately and rigorously? 

Reviewer #1: Yes

4. Have the authors made all data underlying the findings in their manuscript fully available?

Reviewer #1: Yes

5. Is the manuscript presented in an intelligible fashion and written in standard English?

Reviewer #1: Yes

6. Review Comments to the Author

Reviewer #1: (No Response)

7. PLOS authors have the option to publish the peer review history of their article (what does this mean?). If published, this will include your full peer review and any attached files.

Reviewer #1: No

---

## [Editor Report · Acceptance letter]

21 May 2021

PONE-D-20-29337R1 

Sleeping pattern and activities of daily living modulate protein expression in AMD 

Dear Dr. Anand:

I'm pleased to inform you that your manuscript has been deemed suitable for publication in PLOS ONE. Congratulations! Your manuscript is now with our production department. 

Kind regards, 

on behalf of

Dr. Alfred S Lewin 

Section Editor

PLOS ONE